# Real-Time Target Detection System for Intelligent Vehicles Based on Multi-Source Data Fusion

**DOI:** 10.3390/s23041823

**Published:** 2023-02-06

**Authors:** Junyi Zou, Hongyi Zheng, Feng Wang

**Affiliations:** School of Automotive and Traffic Engineering, Wuhan University of Science and Technology, Wuhan 430065, China

**Keywords:** machine vision, millimeter-wave radar, multi-source data fusion, YOLOv5 algorithm, target detection

## Abstract

To improve the identification accuracy of target detection for intelligent vehicles, a real-time target detection system based on the multi-source fusion method is proposed. Based on the ROS melodic software development environment and the NVIDIA Xavier hardware development platform, this system integrates sensing devices such as millimeter-wave radar and camera, and it can realize functions such as real-time target detection and tracking. At first, the image data can be processed by the You Only Look Once v5 network, which can increase the speed and accuracy of identification; secondly, the millimeter-wave radar data are processed to provide a more accurate distance and velocity of the targets. Meanwhile, in order to improve the accuracy of the system, the sensor fusion method is used. The radar point cloud is projected onto the image, then through space-time synchronization, region of interest (ROI) identification, and data association, the target-tracking information is presented. At last, field tests of the system are conducted, the results of which indicate that the system has a more accurate recognition effect and scene adaptation ability in complex scenes.

## 1. Introduction

Current sensors commonly used for environment sensing in smart cars are industrial cameras, LIDAR, RADAR (millimeter-wave radar), inertial measurement unit (IMU), a global positioning system (GPS), etc. [1,2,3,4,5]. These sensing systems not only collect the information to be sensed in the environment, but also process the corresponding signals. For example, artificial intelligence approaches are widely used to process the environmental information extracted from vehicle-mounted cameras. To continuously improve its processing speed and accuracy, new neural networks such as DNN and FRCNN are continuously applied [6,7]. Moreover, deep learning and semantic segmentation methods based on visual data are widely used in various traffic scenarios. For instance, in terms of target detection, the advantages of various road target detection algorithms are compared in Reference [8], and the algorithms’ applicable objects and applicable scenarios are explained in detail. A pedestrian detection algorithm for complex scenes is proposed in Reference [9], and an algorithm for the appearance and distance of the measurement to be inspected is proposed in Reference [10]. On the basis of target detection, Reference [11] then introduces a target-tracking algorithm based on convolutional neural networks. After experiments, it is proved that it can complete the track management and trajectory tracking of quantitative targets based on monocular data. In terms of pavement disease detection, a detection method based on a combined semantic segmentation and edge detection model is introduced in Reference [12], which effectively improves the detection accuracy. A perception method based on GPR investigation and improved mask R-CNN is introduced in Reference [13], which achieves accurate detection of vertical joints inside asphalt pavements.

Radar systems are also common in-vehicle sensor sensing systems. Automotive radar systems are responsible for the detection of objects and obstacles, their position, and their speed relative to the vehicle [14]. However, each sensor of the environmental sensing system can sense environmental information individually, but the single device has high limitations, large errors, and different dimensions of collected information [15], which makes it difficult to achieve the detection of targets in complex scenarios [16]. With the continuous development of intelligent networked vehicle technology, the field of intelligent driving has higher requirements for the precision, accuracy, and real-time performance of sensor-perceived information [17]. To satisfy the above requirements, the theory of data fusion sensing based on two or more sensors is proposed [18,19].

Since different sensors have different properties, fusion schemes between LIDAR, RADAR, and cameras are more frequently proposed for solving vehicle and pedestrian perception problems. A fusion sensing system based on an IR camera and LIDAR was proposed in Reference [20] to solve the problem of sensing under different lighting conditions. In Reference [21], a fused camera and LIDAR sensing system optimized for hardware is proposed. The above fusion perception solutions are based on LIDAR and a camera, but considering that LIDAR is expensive and its recognition accuracy is affected by the lighting conditions in the environment similar to a camera, this paper chooses the fusion perception solution of millimeter-wave radar and a camera.

The fusion of millimeter-wave radar and camera sensing solutions has been initially applied in the field of intelligent driving. In the field of road target perception, the fusion of data collected by in-vehicle cameras and millimeter-wave radar for the perception of vehicles and people in different environments is presented in References [22,23,24,25,26,27], respectively, in the offline case. Offline here means that the sensors collect the data first and subsequently perform the data fusion analysis work on the computer. A fusion sensing algorithm with radar as the primary sensor and camera as the secondary sensor was proposed in Reference [17] and verified experimentally to prove that the method has a high accuracy rate. In the field of target tracking and track management, a millimeter-wave radar and camera fusion sensing scheme based on a polar coordinate system is proposed in Reference [28], which can effectively improve vehicle tracking accuracy and reduce computational costs. In Reference [29], a fused sensing algorithm for distributed millimeter-wave radar and camera is proposed, which is capable of accomplishing track association between MMW radar and camera, and the fused track state estimation method has excellent performance. A DNN-LSTM-based fusion tracking method is proposed in Reference [30], and its robustness and accuracy are demonstrated. Combining the above two related studies, this paper proposes a real-time fusion sensing system of millimeter-wave radar and camera based on the NVIDIA Xavier hardware system. It is a real-time system and easy to install in the car, and can be directly connected to the car ECU for other secondary development work about the smart driving field subsequently. To further ensure the accuracy of the whole fusion perception system, target tracking is performed by extending the Kalman filter to further improve the accuracy of the perception system and reduce the false detection rate.

In addition, this paper has also made corresponding enhancements in the system operation aspects. Neural network recognition methods are often used for processing visual data and are widely used in the field of vehicular traffic. A neural network based on multi-scale feature fusion was proposed in Reference [31]. This neural network is trained to combine the different features of two scanned images and thus identify the cracks in the pavement. As mentioned above, YOLO, as a neural network, is widely used in the field of intelligent transportation. For target detection, Reference [32] and Reference [33] used, respectively, YOLOv5s and YOlOv3 networks for the recognition of traffic signals and traffic signs. Among them, Reference [32] introduced the AlexNet network to further judge the recognition results of YOLO, which effectively improves the system recognition efficiency. Reference [34], on the other hand, implements the control of driverless vehicles based on the tracking detection results of the YOLOv3 network for pedestrians. In terms of pavement disease detection, a YOLOX-based pavement disease detection method was proposed in Reference [35], which effectively solves the problem of slow identification by traditional methods. To improve the comfort of driverless vehicles, a pavement disease detection system based on the Scaled-YOLOv4 detection framework was proposed in Reference [36], and the recognition results can be used in the subsequent engineering related to vehicle control. In terms of improving accuracy, a self-supervised learning network based on “SSL YOLOv4” was proposed to solve the problem of manual labeling of data while ensuring the recognition accuracy in Reference [37]. In Reference [38], a detection recognition algorithm based on improved RES-YOLO is proposed, which significantly improves the average recognition accuracy for multi-target vehicles. In this paper, the feature pyramid network of the YOLOv5 network is improved to adapt to the hardware characteristics of the Xavier mobile processing platform, while taking into account the network size and recognition accuracy. The input of this pyramid network is changed from three feature layers to four feature layers and trained with the selected COCO dataset. The network is experimentally proven to have a higher detection rate and a lower miss detection rate.

This paper is conceived as follows, Section 2 describes the system’s framework and working principle. Section 3 presents the development environment and the components. Section 4 introduces the data pre-processing of the different sensors. Section 5 shows the detailed fusion process. Section 6 demonstrates the field tests of the system.

## 2. System Design

The proposed multi-source data fusion-based vehicle target detection system (referred to as target detection system) takes the approach of fusing millimeter-wave radar data and visual recognition results, and its target detection process is as follows:(1)Information acquisition: The jointly calibrated millimeter-wave radar and camera detect the targets in their working area; receive the measured velocity, relative position, azimuth, and other information; and release them in ROS space after pre-processing.(2)Temporal synchronization: Because the data format and transmission period of the two sensors are different, without temporal alignment, it is impossible to obtain the data of the two sensors for the same measurement at the same time, and thus it is impossible to start fusion detection. Because the two sensors are installed in different frontal positions, without spatial alignment, it is impossible to describe the spatial indicators of the same measurement in a unified coordinate system (body coordinate system in this paper). Two sensor data frames with the same timestamp are combined into one data frame and published to the ROS space as the temporal synchronization result. Spatial synchronization, on the other hand, projects the target coordinate points detected by the millimeter-wave radar into the camera coordinate system based on the written sensor joint calibration data and publishes them to the ROS space as the spatial synchronization result.(3)Fusion decision: Based on the results of spatio-temporal synchronization, the detection area obtained by visual inspection is the main focus, combined with the data detected by millimeter-wave radar to further improve the accuracy of the detection frame and complete the detection process.

The target detection system is mainly composed of vehicle platform, millimeter-wave radar, industrial camera, and intelligent driving domain controller, and its detection process is shown in Figure 1.

Object mode is an operating mode that comes with millimeter-wave radar, and the filtering algorithm built into its controller can filter out some of the stray points. Filter is a pre-processing method proposed in this paper, and its backbone is shown in Filter processing algorithm flow. ID is the number that the ROS environment puts on the data when they are received, and the timestamp is the time when the data are received. The extended Kalman filter (EKF) is used for target tracking and trajectory. The EKF is then used for target tracking and track management.

The functions of each part of the target detection system are as follows:(1)On-board platform: This section is used to mount the sensors, domain controller, and display. After the sensors are installed, the external parameters of the sensors are jointly calibrated according to their installation position.(2)Sensor: It senses the position, size, and azimuth of the target to be detected in the environment and uploads the data to the controller using CAN(Controller Area Network) communication and FPDlink communication.(3)Domain controller: Deploy a real-time target detection program to complete sensor data pre-processing, data fusion, and target detection on the domain controller, and publish the results.

## 3. System Composition

### 3.1. Hardware System

The hardware components of this target detection system mainly include vehicle platform, controller, sensing equipment, communication module and regulated power supply, etc. The names of the devices and their functions are introduced as shown in Table 1.

A schematic diagram of the hardware platform components and the information flow between the main components is shown in Figure 2.

### 3.2. Software System

The software environment of this system is mainly based on an Ubuntu 18.04 LTS operating system on the NVIDIA Xavier AGX development board with ROS melodic and common NVIDIA GPU high-performance image processing software. Ubuntu 18.04 is the most stable and widely used open-source operating system, and Jetpack is NVIDIA’s software management system specifically for this hardware control platform. The ROS software is a common upper computer software in the field of robot control. Further development is on the basis of these functions to complete space-time synchronization, data fusion, and other functions efficiently and conveniently. Cuda is a computing framework introduced by NVIDIA specifically for high-performance CPU computing, which is conducive to solving high-frequency computing problems. The yolov5 model requires the deployment of PyTorch based on the Cuda framework. PyTorch is for deep learning, and OpenCV is an open-source library that is widely used for processing camera images. The specific software environment is shown in Table 2.

For the problem of transferring software data streams, this paper adopts a data communication method based on ROS messages. The system is based on the ROS environment, and the required ROS topics and their contents are defined. The data for the main links of data reading, pre-processing, space-time synchronization, target detection, and tracking can be obtained directly by subscription in the ROS space.

Finally, the UI interface subscribes to the detection results, visualizes the message, and then projects it to the corresponding location to complete all the functions of the detection system.

## 4. Data Processing

### 4.1. Visual Data Processing Based on Improved YOLO Algorithm

The vision detection part of this paper uses an improved YOLOv5 detection algorithm. The YOLO algorithm is a machine vision target detection algorithm that balances real-time and accuracy. The original YOLO algorithm was proposed by Divvla et al. in 2016 [39]. The original YOLO network had some drawbacks in terms of detection accuracy and long recognition time [40].

To address the problems of insufficient detection accuracy and long detection time, this paper uses the latest YOLOv5 target detection network for the purpose of improving the real-time detection speed. The network mainly consists of three parts: the Backbone, the feature pyramid network (FPN), and the classifier and regressor (YOLO Head). The Backbone uses the CSPdarknet network to extract features from the input images, resulting in four feature layers of different scales, Parts 1–4. Part 4 is the smallest scale layer for detecting small targets, while Part 1 is the opposite. The data in Parts 2–4 are fed into the FPN layer and continue to extract features, based on which the other feature layers are upsampled and downsampled, and CSP features are processed. After the scales are the same, these data are fused, which in turn output three final YOLO Head detectors for detection.

The system further improves the detection accuracy by changing the input part of the feature pyramid network (FPN) layer from the original four layers to three layers, and its network structure is shown in Figure 3.

In order to verify the recognition effect of the improved network, this paper conducts a comparative test on the detection effect of the improved network based on the COCO dataset. The full name of MS COCO is Microsoft Common Objects in Context, which originated from the Microsoft COCO dataset funded by Microsoft in 2014 for labeling, and is one of the most common datasets currently used in the intelligent driving domain. This dataset contains the training set for the required scenarios, as shown in the table above, with a rich training set and a corresponding test set. The training set and test set composed from this dataset can effectively demonstrate the performance improvement of the improved YOLOv5 network. A total of 4000 images were selected from the COCO dataset to form the test set, and the scenes covered different road environments and different vehicle conditions under various weather conditions. In order to test its training effect, we selected 1000 images in the same structure as a test set to check its recognition effect. The specific division of the data set is shown in Table 3a, and the results are shown in Table 3b.

### 4.2. Millimeter-Wave Radar Data Processing

In this paper, we use Continental ARS408-21 mm wave radar, which has good software and hardware adaptability and can use the CAN network to configure radar parameters and release information related to the detected measurements [41]. This type of radar has Object mode and Cluster mode, and in this paper, Object mode with certain pre-processing functions is chosen. In Object mode, the radar outputs data such as measured distance, velocity, and mass information contained in Cluster mode, but also performs some hierarchical clustering of detected points, which can output data with fewer clutter points including collision avoidance information. A comparison of the point cloud distribution between the two modes is shown in Figure 4.

The top left is the radar point cloud map of a complex environment in Cluster mode, and the top right is the radar point cloud map of a complex environment in Object mode; the bottom left is the radar point cloud map of an open area in Cluster mode, and the bottom right is the radar point cloud map of open area in Object mode. It can be seen that the filtering algorithm has significantly improved the radar point cloud accuracy in different scenarios.

In addition, in this paper, the data output in Object mode is processed for the application environment of the target detection system to further improve the data accuracy. The processing algorithm flow is shown in Figure 5.

## 5. The Key Technologies in Data Fusion

### 5.1. Time Synchronization

Temporal synchronization of data fusion is when the data collected by two sensors at the same moment are packaged and published [42]. Spatial synchronization of data fusion is divided into soft and hard synchronization. Hard synchronization refers to the homemade multi-sensor time triggers and the use of the triggers to unify the sensor data to be fused in terms of frequency. Soft synchronization, on the other hand, refers to the processing of the received data from different sensors according to the program and system time, thus achieving temporal synchronization of the data at the software level.

The Continental ARS408 mm wave radar and the Senyun industrial camera used in this system are less compatible with hard synchronization, so this system uses a program based on the ROS system to implement millimeter-wave-vision soft synchronization. The two sensor data forms are shown in Figure 6.

As can be seen from Figure 6, the millimeter-wave radar and camera send data to the host computer at regular intervals, and this interval is the output period. The output period of the camera data is not fixed, and its time is between 77 ms and 96 ms. The camera data output period is long, and the interval is variable; the time synchronization is based on the camera data, and the nearest millimeter-wave radar data are matched with the camera data. The time synchronization process is shown in Figure 7.

### 5.2. Space Synchronization

The spatial synchronization of data fusion is the transfer of two sensor data into one coordinate system by matrix transformation [43]. In this system, there are a total of four coordinate systems, such as body coordinate system, camera coordinate system, image coordinate system (ignoring distortion), and millimeter-wave radar coordinate system, whose positions and positive directions are shown in Figure 8.

In the target detection system, if the camera is considered as a coordinate system origin, then it can form a camera coordinate system in the three-dimensional space, as shown in Figure 7, and the points in this coordinate system have three dimensions of coordinates. Because the body coordinate system and the camera coordinate system are in the same three-dimensional space, the points between the two coordinate systems can be obtained by translation, rotation, and other operations. The image coordinate system is the coordinate system that exists for the pictures taken by the camera. It is a two-dimensional plane coordinate system with the center of the image as the origin. The coordinate system is in projection with the camera coordinate system, and the conversion of the points between the two coordinate systems is based on the internal and external reference calibration data of the camera. The conversion formula for the conversion from the measurement points in the body coordinate system to the measurement points in the image coordinate system is shown in Equation (1).
(1)$$\left[\begin{array}{c} x\\ y\\ 1 \end{array}\right]=\left[\begin{array}{cccc} f_{x} & 0 & c_{x} & 0\\ 0 & f_{y} & c_{y} & 0\\ 1 & 0 & 1 & 0 \end{array}\right]\left[\begin{array}{cc} R & T\\ 0^{T} & 1 \end{array}\right]\left[\begin{array}{c} X\\ Y\\ Z\\ 1 \end{array}\right]=M_{1}M_{2}X$$

In Equation (1), x,y are the image coordinate system coordinates; $f_{x}$, $f_{y}$, $c_{x}$, $c_{y}$ are the camera internal reference matrix parameters, $M_{1}$ is the camera internal reference matrix; R, T are the camera external reference matrices, R matrix is the orthogonal matrix, representing the rotation around the axis, T matrix represents the axial offset. $M_{2}$ is the external reference matrix; X, Y, Z are the measurement coordinates in the body coordinate system.

The measurement detected by the millimeter-wave radar is in the radar coordinate system, and in order to perform spatial synchronization, the measurement coordinates need to be converted to the body coordinate system. As shown in Figure 6, the millimeter-wave radar and the body coordinate system are in the same three-dimensional space, so the transformation from radar coordinate system measurement to body coordinate system measurement can be realized by rotation and translation, and the conversion formula is shown in Equation (2).
(2)$$\left\{\begin{array}{c} x=R\times \sin\alpha \\ y=H+R\times \cos\alpha \end{array}\right.$$

In Equation (2), H is the distance from the projection point of the radar coordinate system origin on the XOY plane of the body coordinate system to the origin of the body coordinate system, R is the distance from the projection point of the obstacle on the XOY plane to the projection point of the radar coordinate system origin on the XOY plane of the body coordinate system, and α is its angle with the y-axis.

Equations (1) and (2) complete the conversion of the image coordinate system, the radar coordinate system to body coordinate system, respectively, and the relationship from the radar coordinate system to the visual coordinate system is obtained by matrix transformation of the two (Equation (3)). So far, the spatial alignment from radar data projection to image data has been completed, and the flow chart is shown in Figure 9.
(3)$$\left[\begin{array}{c} u\\ v\\ 1 \end{array}\right]=\left[\begin{array}{cccc} f_{x} & 0 & c_{x} & 0\\ 0 & f_{y} & c_{y} & 0\\ 1 & 0 & 1 & 0 \end{array}\right]\left[\begin{array}{cc} R & T\\ 0^{T} & 1 \end{array}\right]\left[\begin{array}{c} R\times \sin\alpha \\ H+R\times \cos\alpha \\ 0\\ 1 \end{array}\right]$$

### 5.3. Data Fusion

Compared to the region of interest (ROI) generated by millimeter-wave radar [44] used by Yanping Hu et al. in this paper, the improved YOLOv5 network is chosen, and the visual detection frame accuracy has been greatly improved compared to the earlier version of the YOLO network; therefore, the visual detection results are chosen to be directly adopted in this paper.

Due to the extremely metal-sensitive nature of millimeter-wave radar and the fact that the location of the millimeter-wave contact with the detected object is not fixed, it is easy to produce drift of the measurement points [45]. Therefore, when calculating the distance between the measurement and the on-board experimental platform and the detection probability of the millimeter-wave on the measurement, the average value of all millimeter-wave point data in the detection frame is used as the detection result of the millimeter-wave radar.

When both sensors detect the object, the final detection result is output according to Equation (4).
(4)$$\left\{\begin{array}{c} W=P_{C}\times 0.8+P_{R}\times 0.2\\ Y=P_{C}\times 0.2+P_{R}\times 0.8\\ X=P_{C}\times 0.2+P_{R}\times 0.8 \end{array}\right.$$
where W is the measured detection probability, X is the horizontal distance, Y is the vertical distance, and $P_{C},\;P_{R}$ are the corresponding camera data and radar data, respectively. In Equation (4), we refer to the weighted determination method in Reference [46] on determining the presence or absence of fusion targets, and propose the output method of the measurement metrics based on the weighted data of the two sensors in Equation (4).

The output data mainly include the judgment of the measurement category and the value of the measured horizontal longitudinal distance. Firstly, since there are only two sensors, the coefficients need to add up to 1. Secondly, the two sensors have different hardware sensing principles, and the sensing accuracy of different indicators is relatively high or low. For engineering practice and prior experience, we believe that visual judgment has a higher accuracy rate in target detection, and radar has a unique advantage in distance perception. Therefore, we set the weight of camera data in W to 0.8 and radar data to 0.2, and the weight of camera data in X and Y to 0.2 and radar data to 0.8. 

The rest of the cases are output according to the fusion decision shown in Table 4.

### 5.4. Target Detection and Tracking

Due to complex scenes or changes in vehicle conditions, the visual detection system and the millimeter-wave radar may lose the target for a short time, and the fused sensory information at this moment will be lost if no relevant processing is performed at this time. To solve this problem, this paper selects the extended Kalman filter (EKF) for fused data points for track management and tracking, which is widely used for tracking and managing target tracks because of its ability to combine sensory information and prediction results to predict the future state of the system [47].

In this paper, the mobile platform deployed by the real-time detection system is the coordinate origin, the forward direction is the y-direction, and the lateral direction is the x-direction, then the state vector of the target measurement is:(5)$$X={\left(x,y,v_{x},v_{y}\right)}^{T}$$

The x, y, $v_{x},v_{y}$ in Equation (5) are the position coordinates and velocity of the target measurement.

The equation of state and the equation of observation for the EKF are, respectively
(6)$$\left\{\begin{array}{c} x_{k}=f\left(x_{k-1},u_{k}\right)+w_{k}\\ z_{k}=h\left(x_{k}\right)+v_{k} \end{array}\right.$$

In Equation (6), $x_{k},x_{k-1}$ are the state vectors at the moment of measurement k and k −1, respectively; $u_{k}$ is the control vector, the system does not use the control strategy in the fusion stage, the item is 0; $w_{k}$ is the process evolution noise (usually Gaussian noise); $z_{k}$ is the sensor-perceived measurement observation vector; $f\left(x\right),h\left(x\right)$ are the system nonlinear state function and measurement function, respectively; $v_{k}$ is the measurement noise (usually Gaussian noise). The above are the data values at the moment of k.

According to the EKF, the predicted state of the measurement target at the kth moment is
(7)$$X_{k}=f\left(x_{k-1},u_{k}\right)$$

$X_{k}$ in Equation (7) is the predicted value measured at the kth moment.

In addition, the prediction error covariance can be obtained as:(8)$$P_{k}=F_{k}P_{k-1}F_{k}^{T}+Q_{k}$$

In Equation (8), $P_{k}$ is the prediction error covariance measured at moment k, F is the accordance ratio matrix of $f\left(x\right)$, and $Q_{k}$ is the covariance matrix representing the prediction state.

From Equations (6) to (8), the update equation can be obtained:(9)$$\left\{\begin{array}{c} K^{\prime }=P_{k}H_{k}^{T}{\left(H_{k}P_{k}H_{k}^{T}+R_{k}\right)}^{-1}\\ P_{k}^{\prime }=P_{k}-K^{\prime }H_{k}P_{k} \end{array}\right.$$

In Equation (9), K is the Kalman gain, $H_{k}$ is the transformation matrix that maps the state vector $x_{k}$ to the space where the measurement value $z_{k}$ is located; $R_{k}$ is the covariance array of Gaussian noise of the measurement value, representing the measurement error of the sensor, $K^{\prime },X_{k}^{\prime },P_{k}^{\prime }$ are the updated values.

The final state matrix of the measurement is obtained as
(10)$$X_{k}^{\prime }=X_{k}+K^{\prime }\left(z_{k}-h\left(X_{k}\right)\right)$$

Equations (7)–(10) combine the observed value at moment *k* − 1 and the predicted value at moment *k* to calculate the position estimate at moment *k*. At the same time, the parameters such as the prediction error covariance are corrected in real time by combining the observed value at moment *k*, and the state matrix of the measurement is updated to realize the cyclic proceeding of the algorithm. The above process achieves the same measurement of past and present state matching.

In summary, the flow of the fusion detection algorithm used in this system is shown in Figure 10.

## 6. Equipment Commissioning and Real Vehicle Test

### 6.1. Equipment Calibration

#### 6.1.1. Internal Reference Calibration

The camera internal reference calibration of the target detection system mentioned in this paper is performed using MATLAB’s camera calibration toolbox. The calibration plate is shown in Figure 11a, and the spatial position of the 20 calibration plate photos with respect to the camera is shown in Figure 11b, from which the resulting internal reference calibration parameters are given by Equation (11).
(11)$$\left(\begin{array}{ccc} 3953.92182439858 & 0 & 1009.14636095190\\ 0 & 3958.02267342530 & 537.408452046159\\ 0 & 0 & 1 \end{array}\right)\;$$

#### 6.1.2. External Reference Calibration

In this paper, an open-source offline program is used for the external reference calibration. The time-synchronized millimeter-wave point cloud data and visual image data are input into this program, and the radar point cloud positions are adjusted to coincide with the measurements on the visual image, and finally the axial offsets of the millimeter-wave point cloud data in X, Y, and Z and the rotational offsets with X, Y, and Z as the rotation axes, respectively, are recorded, and these six parameters represent the external parameters of this system [48]. The final calibration parameters are shown in Equation (12).
(12)$${\left(\begin{array}{ccc} 0 & 80 & 0 \end{array}\;\;\;\;\;\begin{array}{ccc} 0.02 & 0 & 0 \end{array}\right)}^{T}\;$$

### 6.2. Fusion Effect

In this system, for the convenience of external parameter calibration and equipment deployment, the millimeter-wave radar and camera are chosen to be deployed on the robot rail experimental stand, installed on the vehicle experimental platform, and then the experimental platform is deployed on the modified electric patrol car, and the final arrangement effect is shown in Figure 12.

To visualize the fusion effect, the millimeter-wave radar points that have been spatio-temporally synchronized are placed directly on the screen in this paper, and the horizontal coordinates and distances of the measurements are calculated, and the effect is shown in Figure 13. “99.0” is the vertical distance of 99.0 m, “x 11.80” is the horizontal distance of 11.0 m, and the purple point is the millimeter-wave radar measurement point.

To test the effectiveness of the system in detecting vehicles and pedestrians at noon and evening, the system was deployed on a mobile platform, and experiments were conducted on campus scenes, and the vehicle recognition results are shown in Figure 14. The blue box in the figure is the recognition frame of the improved YOLOv5 visual recognition network detection used in this paper, the pink dots are the measurement points detected by the millimeter-wave radar, x is the horizontal distance of the measurement point from the millimeter-wave radar, and y is the longitudinal distance of the measurement point from the millimeter-wave radar. The fusion detection is considered successful when the millimeter-wave radar point and the vision frame appear on the measurement object at the same time.

In this paper, relevant tests were also conducted in scenes with high pedestrian flow, and the vehicle effect is shown in Figure 15. In the face of complex scenes, the system has a high accuracy of recognition precision and a low rate of missed detection.

In order to verify the gap between the detection effect of the sensor fusion algorithm and that of a single sensor, 300 sets of data after spatio-temporal alignment were selected and processed with single vision detection, single radar detection, and fusion detection, respectively, and Table 5 shows the processing results of the three methods.

The results in Table 5 show that the fusion target detection algorithm used in this system has a higher detection rate than the traditional method for road pedestrian and vehicle identification, and the accuracy rate under fusion detection is higher as seen in Figure 13 and Figure 14. At the same time, this system also combines the advantages of millimeter-wave radar to make up for the shortcomings of visual detection which makes it difficult to output the measured distance information, thus improving the scientific and applicability of the fusion detection algorithm.

## 7. Conclusions

In this paper, we design and implement an on-board real-time multi-target detection system, which fuses the filtered pre-processed millimeter-wave radar data and the visual data detected by YOLO for target-level data fusion, and then adopts the corresponding fusion strategy for the output of the relevant measurement targets. In addition, in order to reduce the output of pseudo-targets before the fusion results are output, this paper adopts the method of the extended Kalman filter to manage the fused measured trajectories. Compared with the traditional visual recognition detection algorithm, the proposed system in this paper outputs the measurement results with the distance information simultaneously, which improves the applicability of the system.

The experimental results show that the target detection method has a high detection rate and a low false detection rate, and the fusion of visual detection data and millimeter-wave radar data is effectively accomplished in a variety of scenarios, and the target detection effect is better than that of single-sensor detection. While performing the detection, this system can output the distance of the obstacle and the position relative to the vehicle, which improves the richness and accuracy of the detection data.

However, problems such as short delay in the fusion procedure and increased missed detection rate of complex scene target recognition were also found in the process of experimentation. In the subsequent experimental research, the target detection fusion algorithm should be further optimized to shorten the system delay, the YOLO network should be trained for more complex environments, and the algorithm structure should be adjusted by repeatedly conducting multi-scene real vehicle detection tests, so as to further improve the real-time performance and complex scene adaptability of the target detection system.

## Figures and Tables

**Figure 1 sensors-23-01823-f001:**
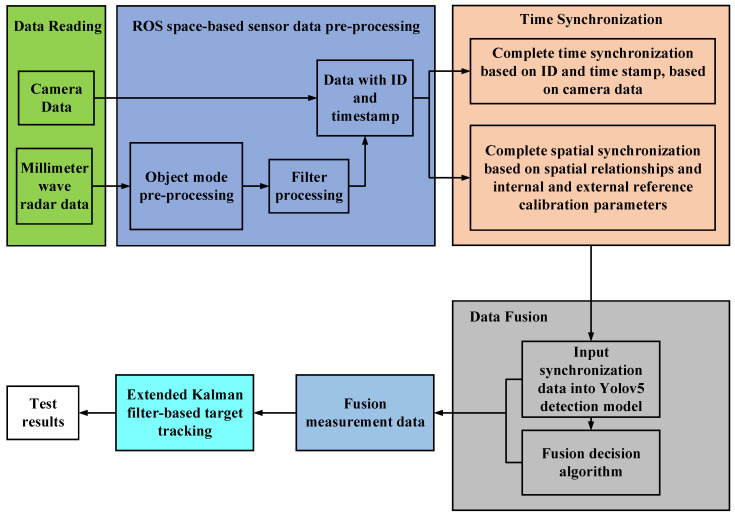
Target detection flow chart. The integration process is described below.

**Figure 2 sensors-23-01823-f002:**
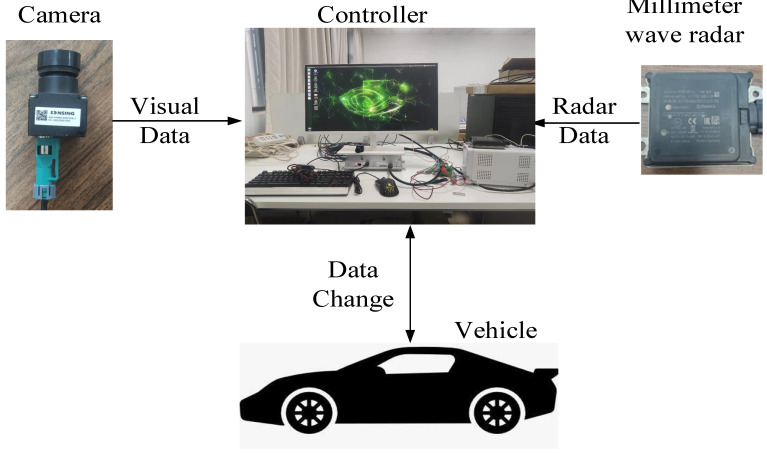
Hardware platform composition diagram.

**Figure 3 sensors-23-01823-f003:**
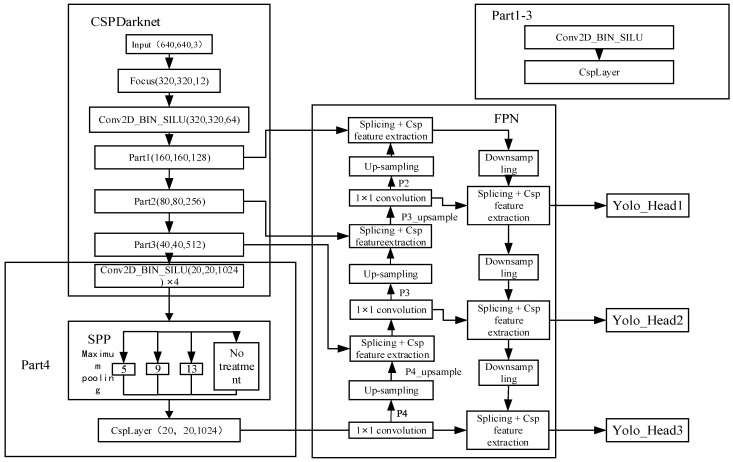
Target detection flow chart.

**Figure 4 sensors-23-01823-f004:**
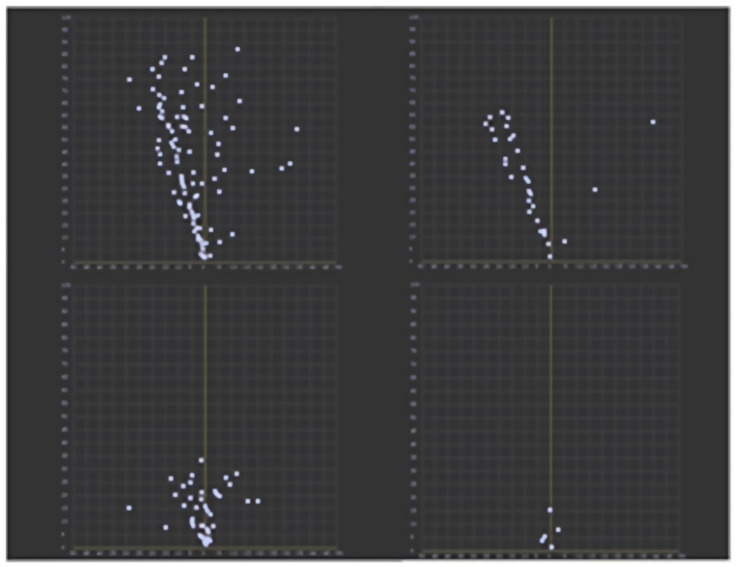
Target detection flow chart.

**Figure 5 sensors-23-01823-f005:**
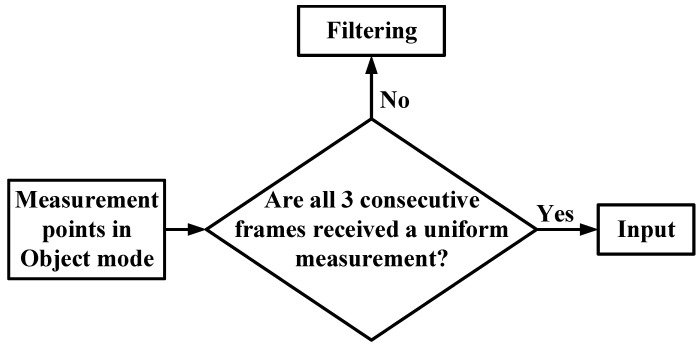
Filter processing algorithm flow.

**Figure 6 sensors-23-01823-f006:**
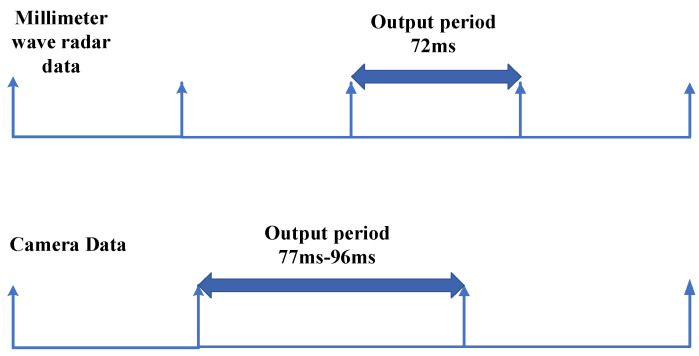
Data form. The axes in Figure 6 are measured in milliseconds.

**Figure 7 sensors-23-01823-f007:**
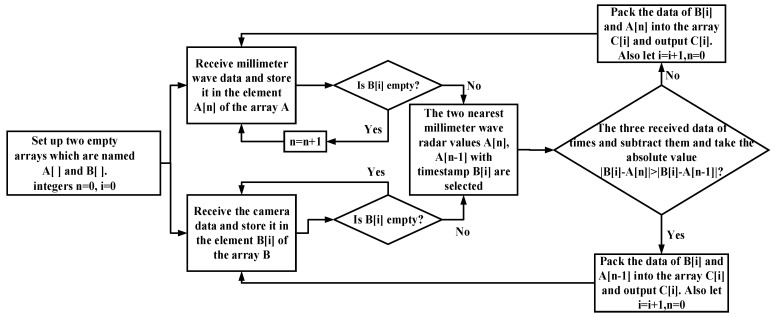
Time synchronization flow chart.

**Figure 8 sensors-23-01823-f008:**
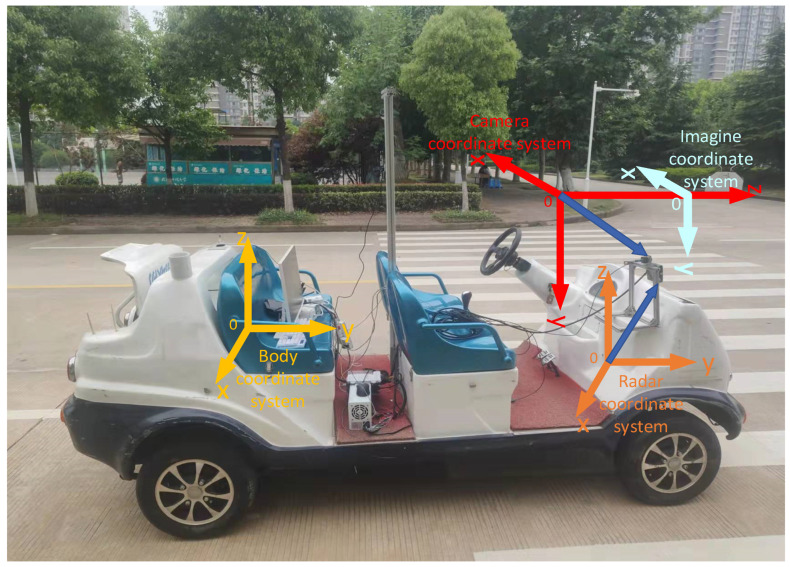
Schematic diagram of the coordinate system of the testing system. The red one is the camera coordinate system, the yellow one is the body coordinate system, the orange one is the millimeter-wave radar coordinate system, and the light-blue one is the image coordinate system.

**Figure 9 sensors-23-01823-f009:**
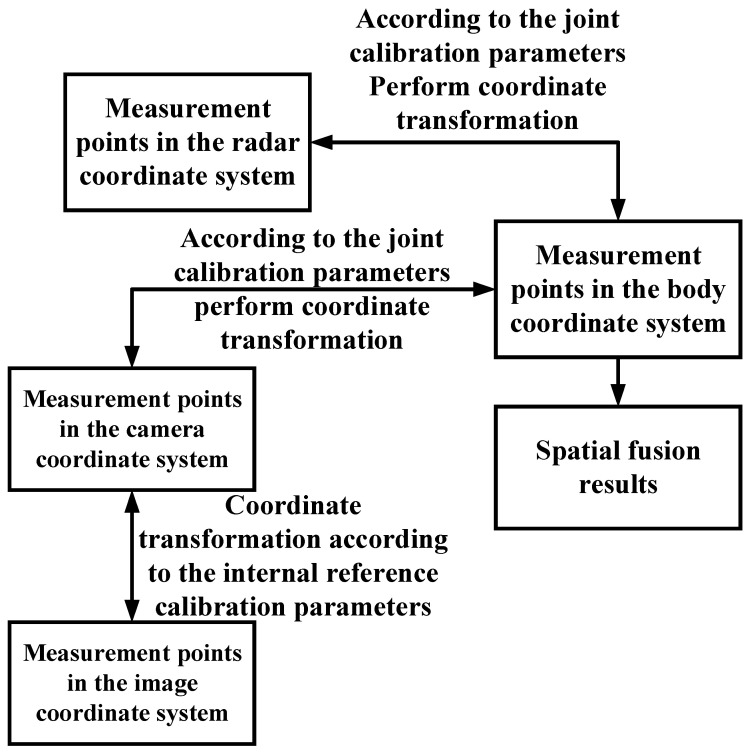
Spatial Alignment Flow Chart.

**Figure 10 sensors-23-01823-f010:**
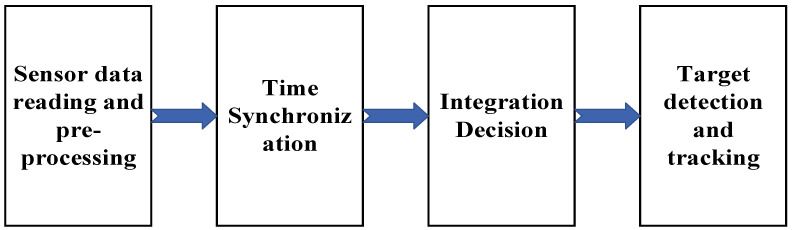
Flow chart of fusion algorithm.

**Figure 11 sensors-23-01823-f011:**
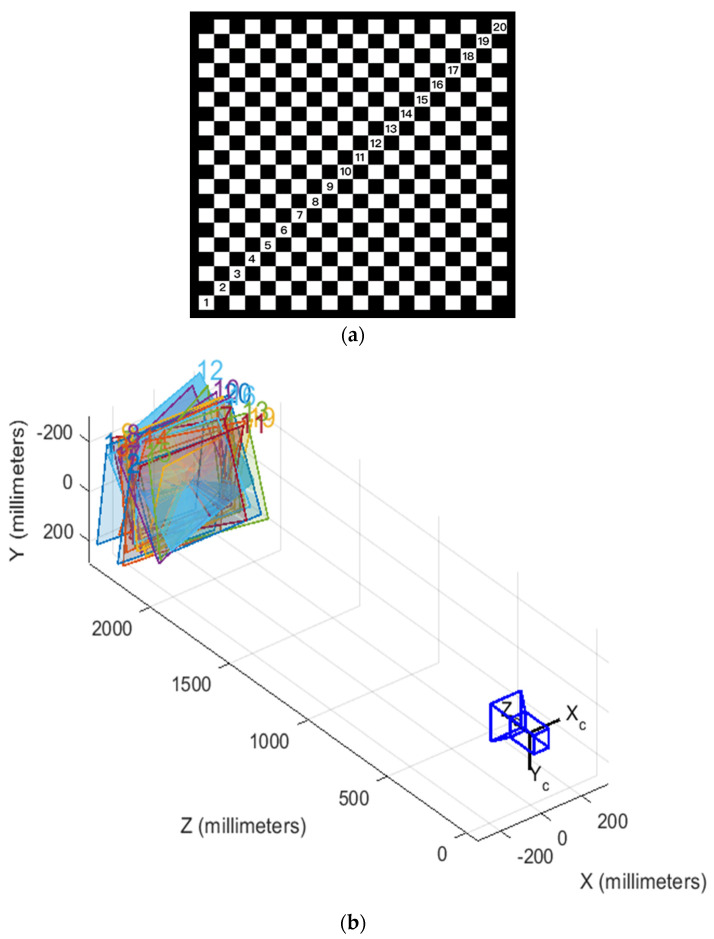
(**a**) 20 × 20, 25 mm calibration plate; (**b**) spatial position of the calibration plate with respect to the camera.

**Figure 12 sensors-23-01823-f012:**
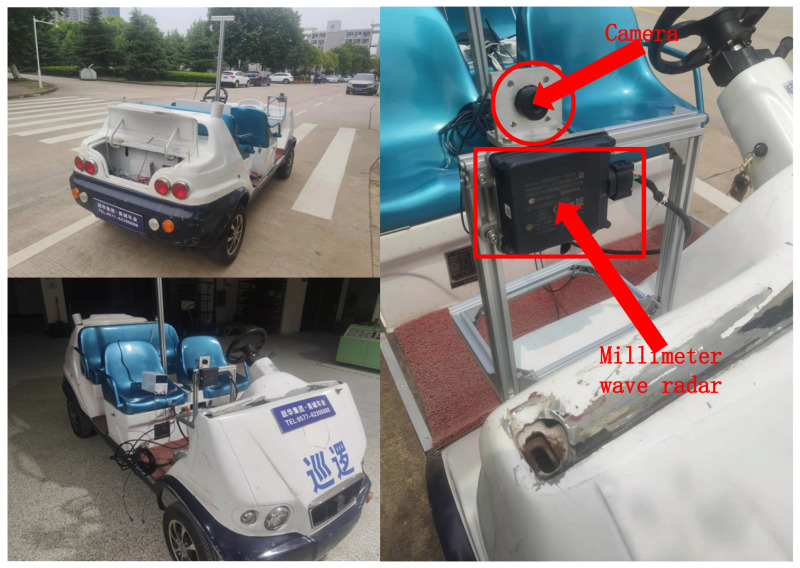
Vehicle-mounted experimental platform.

**Figure 13 sensors-23-01823-f013:**
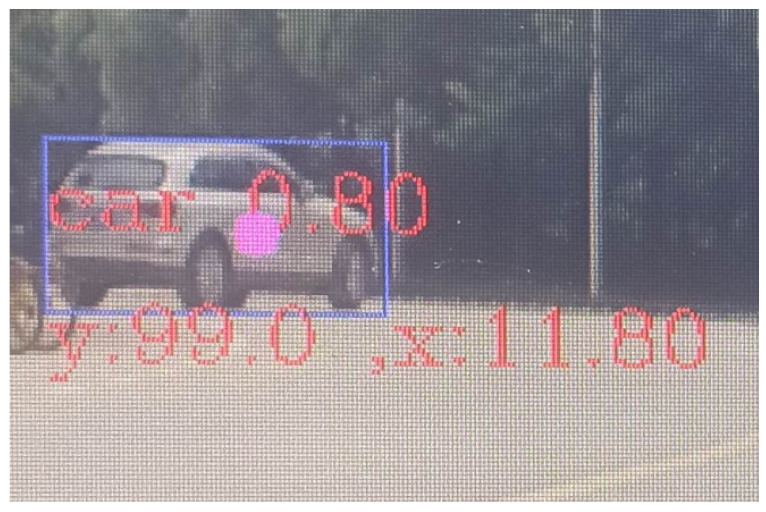
Fusion display effect.

**Figure 14 sensors-23-01823-f014:**
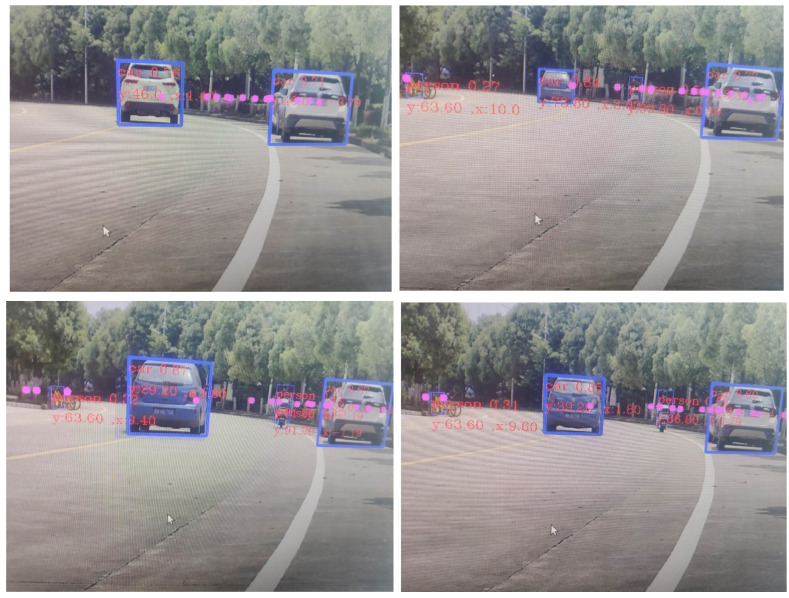
Vehicle recognition effect map.

**Figure 15 sensors-23-01823-f015:**
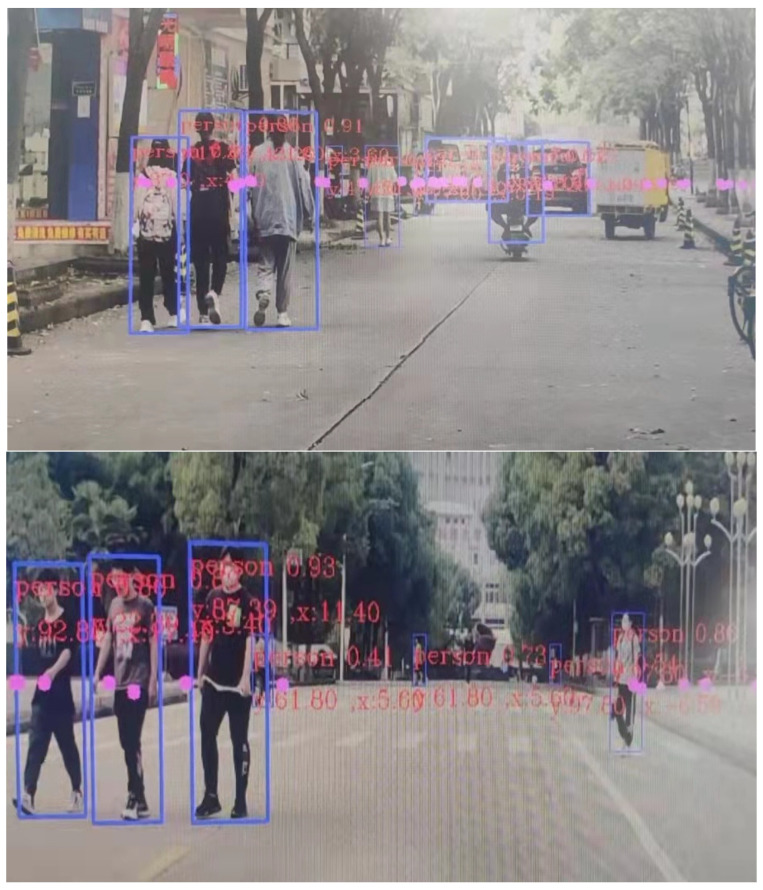
Complex scene recognition effect.

**Table 1 sensors-23-01823-t001:** Target detection system hardware components.

Equipment	Quantity	Function
Senyun Smart Industrial Camera	1	Acquisition of the environment to be measured
Continental ARS-408 mm wave radar	1	Obtain information such as the distance and position of the target to be measured in the environment
Cuckoo Autopilot Domain Controller	1	Deploy programs, receive data, and publish identification results, system power supply
12V DC regulated power supply	1	System power supply
7-inch display	1	Display the recognition effect of the program and related information
Communication Module	1	Data transfer between sensors

**Table 2 sensors-23-01823-t002:** Software environment of the Target detection system.

Software Name	Versions
Ubuntu	18.04 LTS
JetPack	4.4.1
Cuda	10.2.89
Opencv	4.1.1
ROS	Melodic
Python	3.6.9
YOLOv5	PyTorch version

**Table 3 sensors-23-01823-t003:** (**a**) The specific division of the data set. (**b**) Recognition effect comparison results.

**(a)**						
**Training Set**	**Sunny**	**Rain**	**Cloudy**	**Test Set**	**Sunny**	**Rain**
**Daytime open space**	400	300	300	**Daytime open space**	100	75
**Daytime Road Complex**	400	300	300	**Daytime Road Complex**	100	75
**Evening open space**	400	300	300	**Evening open space**	100	75
**Evening Road Complex**	400	300	300	**Evening Road Complex**	100	75
**(b)**						
**Name**	**Model size**	**Image size**	**Detection speed (fps)**	**Average positive inspection rate (%)**	**Recall**	
Improve YOLOv5	168 M	640 × 640	31.9	88.06	80.46	
YOLOv5	90.2 M	640 × 640	32.6	87.21	79.98	
YOLOv3	237 M	640 × 640	16.5	85.54	77.66	

**Table 4 sensors-23-01823-t004:** Integration Strategy.

Camera	Millimeter-Wave Radar	Strategies
Target detected	Target detected	Perform fusion detection
No target detected	Target detected	Output millimeter-wave radar detection data
Target detected	No target detected	Output visual monitoring data
No target detected	No target detected	Judged by the extended Kalman filter

**Table 5 sensors-23-01823-t005:** Comparison of target detection results. The first data in each cell is for identifying vehicles and the second data is for identifying pedestrians.

Detection Method	Total Number of Targets (Vehicles/People)	Positive Inspection	Misdetection	Missing Inspection	Distance Information Error	Accuracy %	Missing Detection Rate %
Single vision inspection	2604 (832/1772)	716/1521	724/1544	116/251	None	86.06	13,94
						85.84	14.16
Single radar detection		631/1323	740/1602	201/449	<2 m	75.84	24.16
						74.66	25.34
Fusion Detection		766/1604	74/172	66/168	<2 m	92.07	7.93
						90.52	9.48

## Data Availability

Not applicable.

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
