# Peer review of "Real-Time Target Detection System for Intelligent Vehicles Based on Multi-Source Data Fusion"

_sensors, 2023, doi:10.3390/s23041823_

Round 1

Reviewer 1 Report

This paper proposed a real-time targets detection algorithm based on radar and camera. However, there are some problems need to be improved.

1.  How much the test set for Table 3 in the Section 4.1? The accuracy and Precision of the detection algorithm should be evaluated according to the indexes of true positives, true negatives, false positives, false negatives.

2. In Fig.8, the image coordinate could be added since the image coordinate and camera coordinate are easy to be confused.

3. For the KALMAN filter in the target tracking, how to match the position date between the history state and the updated state?

4. In Fig.6, what is the output period? what is the meaning of 77-96?

5. Please complain the target detection flow in Figure 4.

6. The title of section 6 should be changed.

7. The clarity of the picture should be improved, such as Figure 14, etc.

8. In Table 5, there are two detection rates for each method, please explain which target corresponds to each detection rate.

9. In the introduction, the literature review seems to be weak with needs to be enhanced with more publications.

Reviewer 3 Report

See attached file. 

Round 2

Reviewer 3 Report

Dear authors, thanks for the quality improvement of the paper. All my requests have been met. Just one last thing: I think the caption of figure 11 can be better written. 
